# A Qualitative Study of Parents’ Conceptualizations on Fever in Children Aged 0 to 12 Years

**DOI:** 10.3390/ijerph16162959

**Published:** 2019-08-16

**Authors:** María Gloria Villarejo-Rodríguez, Beatriz Rodríguez-Martín

**Affiliations:** 1Health Center of Bargas, 45593 Toledo, Spain; 2Social and Health Care Center, University of Castilla-La Mancha, 16007 Cuenca, Spain; 3Faculty of Health Sciences, University of Castilla-La Mancha, Talavera de la Reina, 45600 Toledo, Spain

**Keywords:** perception, grounded theory, fever, children, parents

## Abstract

Many parents experience “fever phobia”, based on misconceptions regarding the repercussions of fever in their children. The aim of this paper was to explore the conceptualizations of parents who are health professionals and parents without health qualifications on childhood fever. This qualitative study was based on grounded theory using a triangulated sample (theoretical sampling and snowball sampling) of parents of children aged 0 to 12 years old who received care for fever in the Emergency Primary Care Services two in Spanish municipalities. Data collection was based on focus groups segmented by gender, place of residence and education. Data analysis followed the constant comparative method and involved a coding process. Results show that independently of the parents’ place of residence or education, their perceptions of fever were somewhat ambivalent, beneficial at times, but also harmful. Parents acknowledged feelings of concern, fear, being overwhelmed, freezing up and relief once the fever was controlled. Health professional parents considered they had an extra responsibility for caring. Finally, parents without health education demanded more information from professionals. These results provide key information for the design of interventions directed at the management of fever in children.

## 1. Introduction

Parents have a certain tendency to consider fever as a disease, instead of a symptom or sign of illness [1]. Although childhood fever is usually a natural defense mechanism, it is common for parents to manifest anxiety regarding the control of fever in their children [2,3]. Schmitt named this parental fear “fever phobia” or phobia of fever, based on the ensemble of poorly realistic concerns parents have on the repercussions of fever in children [4]. 

We know that phobia to fever in parents decreases when health professionals provide parents with appropriate information and the tools to actively care for their children [5,6]. On the other hand, even though social media and web 2.0 tools have an increasing influence on the dissemination and retrieval of medical information [7], few studies have analyzed the quality of online information and its influence on the management of the population’s health [8,9].

Previous research has shown that a person’s educational level can determine their perceptions on the health-disease process, in which the academic level is inversely related to the amount of health-related knowledge and phobia regarding a range of symptoms [10,11,12]. Besides, it is well known that the health culture of parents can influence lifestyle choices and the approach towards illness. Thus, by having more knowledge in health processes, health professionals favor a more preventive attitude and behavior than that found in the general population [13]. To our knowledge, no studies to date have analyzed the differences in the perceptions of fever among parents with or without health education.

The aim of the present study was to explore the conceptualizations on fever of the parents of children aged 0 to 12 years, differentiating between parents who are health professionals and those who are not.

## 2. Methods

This was a qualitative study based on grounded theory. This inductive method was selected as it enables researchers to obtain a theoretical explanation, in this case, by analyzing the conceptualizations of parents on child fever [14].

### 2.1. Sample and Data Collection

A triangulated sample was included. First, a theoretical sample was recruited, guided by the constant comparative method [15,16], which included parents of Spanish nationality with children aged between 0 and 12 years who, during the period of data collection (November 2016–October 2017), had received care at the Primary Care Emergency Services (PCES) of the health clinics of (blinded for peer review) due to a fever and who voluntarily accepted to participate in the study. The study included parents with an academic title in the field of health as well as parents without such training. The exclusion criteria consisted of parents/caregivers of children with chronic cardiac or respiratory illnesses (asthma, pneumonia, bronchiolitis).

Snowball sampling was used to select parents of children who, besides the previous inclusion criteria, were health professionals in the public or private healthcare sector, whether these were actively employed, on leave, unemployed or retired.

The sampling process continued until the data saturation criteria was fulfilled, at which point, to continue to increase the sample would provide repetitive or similar information without providing new data [17]. 

The first contact with potential participants was performed by telephone via the pediatric nurses of the selected health centers, who were the key informants for the participant selection, before three weeks had passed from their PCES consultation.

For data collection, focus groups (FGs) were used, by segmenting the sample into parents/caregivers with and without health education, and residents in rural or urban areas. Also, the sample was segmented by gender due to the possible influence of the same as an inhibitory element reported in previous studies on care [18]. For the organization of the FGs, the criteria of intragroup homogeneity and heterogeneity were considered [19] (Table 1). Data saturation was obtained after eight FGs (57 participants). These were four groups of parents/caregivers without health training, half of these groups resided in a rural area and another half were from an urban area, and four groups of parents who had health training, two for each type of health clinic: i.e. rural and urban. The FGs of fathers took place separately from those of mothers (i.e., these were separated by gender).

The FGs were conducted between January and October 2017. These took place in a calm and private environment and with a mean duration of 50 minutes and including a maximum of ten participants per group (half of these were first-time parents). The FGs were audio recorded after obtaining written consent from participants. These were led by a moderator, who followed a guide of themes that could appear during the sessions, which was refined as the study progressed. Also, an observer monitored the groups. Both researchers had training and experience in qualitative research (MGVR and MJ (collaborator)). 

### 2.2. Data Analysis

The recorded material was anonymized (assigning capital and lowercase letters to each of the participants) and transcribed verbatim, line by line (using F4 software). The transcriptions were sent by email to the participants to verify the accuracy of the transcriptions.

Two researchers independently performed the data analysis, after which they discussed the results to obtain consensus (MGVR and BRM). This analysis was based on the constant comparative method and on processes of open, axial and selective coding, supported by Atlas.ti 7.0. software (ATLAS.ti Scientific Software Development GmbH, Berlin, Germany) [15,17,20]. 

For the systematic examination of the text, the themes were identified and classified by coding the content of the discourse. After the preliminary reading, a global vision of the useful ideas was obtained for the subsequent analysis, obtaining the first memos (annotations for the process of analysis, data collection and other instrumental processes) [21]. Each transcription was analyzed before performing the next focus group (FG), in order to evaluate the data saturation [17]. 

The first analytical step, open coding, enabled the extraction of concepts to group these into categories under a common semantic field. This categorized data was then related and regrouped, linking the categories into subcategories to form clearer and more specific explanations of the phenomena related with the study theme (axial coding). Finally, the main categories were integrated into a central category via selective coding and a main theoretical framework was established [22]. The coding process followed a circular, flexible logic during the analysis [14]. 

### 2.3. Validity

This research followed the criteria of The Consolidated Criteria for Reporting Qualitative Studies (COREQ) [23]. The methods used for guaranteeing validity were data triangulation, including participants with different sociodemographic characteristics, and triangulation of data analysis via different researchers. Besides this, the transcriptions of the FGs were returned to the participants for their approval. 

### 2.4. Ethical Considerations

This study was approved by the Clinical Research Ethics Committee of (blinded for peer review) (nº 29 29/02/2016). The confidentiality and anonymity of the data follows the Spanish legislation, i.e., the Organic law of 15/1999, of December 13, on the protection of personal data [24].

## 3. Results

Three main themes explained the conceptualizations of the parents on childhood fever without finding differences according to their place of residence: knowledge and beliefs of parents on fever (Figure 1), parents’ ambivalent feelings regarding their child’s fever (Figure 2) and the need for information on fever perceived by parents (Figure 3). For a better understanding of the results, Appendix A, available online, gathers the emerging categories, subcategories and codes, together with participants’ main verbalizations.

Both the health-educated parents and the non-health-educated parents considered that some children are more “prone” to suffer from fever than others. Also, the appearance of fever was more common during infancy.

The etiological factors of fever perceived by the parents (without differences regarding their previous training) were: attendance to a nursery school, the child’s process of growth, teething, weaning, problems affecting the throat or the ear, colds and runny nose, the excess of heat, or as a reaction to certain vaccines. The cause for fever that was most often mentioned in both groups was the presence of an infection due to virus/bacteria. Also, some parents included chance as a cause for the appearance of fever, whereas others considered the possibility of fever appearing without their being a source for the outbreak. Parents who were health professionals also cited urinary infections as a possible cause of childhood fever. Other causes of fever referred to only in the case of non-health-educated parents were tiredness and stress.

According to the intensity of the fever, both parents who were health professionals and those who were not, classified fever as: low-grade fever (or “normal fever” for some parents without health education), which was considered when the temperature was lower than 38 °C, high fever (up to 40 °C), and very high fever or “fever spikes” (40–41 °C).

Parents from both groups highlighted childhood “immunization” measures as methods for the prevention of fever. This included contagion amongst equals or the administration of vaccines according to the vaccine calendar. The non-health-educated parents also included prophylaxis with probiotics, having a good diet or protecting “weak” organs such as the throat and ears.

The signs and symptoms of fever mentioned by parents of both groups were: overall discomfort, arthralgia, headache, muscle aches, crying, having a hot body, cold or hot hands and feet, rosy cheeks, red ears, glassy eyes, bags under their eyes, “goose bumps”, shivering, lack of appetite, vomiting, and changes affecting the child’s behavior (irritability, sleepiness/malaise, “acting up” or seeking nurture by the mother). 

Additionally, parents, both health professionals and those who were not, mentioned the phenomenon of “tolerance to fever” in children, understood as the adaptation of the child to symptoms related to fever, and the singularity of this process, manifesting and progressing in different manners. The participants of both groups described fever as a process lasting between two or three days, with variations during the day and which are often spontaneously resolved, as a self-limiting process.

Certain parents (with no distinction regarding health education) perceived fever as a positive process, conceptualizing the same as a physiological response, part of the development of the child or as something beneficial, considered to be a defense mechanism of the body or a sign of infection. In contrast, some parents perceived fever as something damaging to health, which could even lead to the death of a child, considering the same as the most extreme complication of fever. Other complications of fever perceived by participants from both groups were febrile convulsions, brain damage or mental problems. Non-health-educated parents also mentioned the possibility of protein denaturation due to the high temperatures.

Most parents (both with and without health training) highlighted feelings of worry and being overwhelmed when they were unable to determine the cause of the fever, or when a febrile episode evolved. Also, in the presence of very high temperatures, or due to the excessive duration of fever or the administration of medication (especially due to the possible adverse effects and the resistance to treatment). These feelings were overlapped with feelings such as fear, also defined by parents as “dire anguish”. In participants of both groups of parents, this was associated with the possibility that their child may have suffered some type of severe illness (e.g., meningitis), or other previously mentioned possible complications of fever. Another recurrent fear manifested by the participants was caring for a child with a febrile episode on one’s own, without the support of another person. Also, part of the parents of both groups stated that the presence of fever in their children caused them feelings of suffering and pity.

Most parents with more than one child, independent of their training, affirmed that their concern was greater with their first child, with these feelings decreasing as their children grew older. Also, they acknowledged that these feelings increased when the fever appeared or persisted at night. Regarding fever assessment and monitoring, the participants in both groups acknowledged experiencing a certain obsession in their behavior when measuring their child’s temperature.

In the case of parents who were health professionals, they acknowledged feelings of distress related with their perceptions of bearing a greater responsibility for caring for their child, (which is assigned to them socially due to their profession). Also, these parents acknowledged that, despite having healthcare training, they felt ‘blocked’ when an episode of fever occurred in their children and, on occasion, they did not know what to do, feeling guilty or like “bad parents” if their actions were incorrect. They also acknowledged feeling foolish in the presence of professional colleagues if they consulted for fever in emergency services.

In contrast to the above statements, there were also participants from both groups who did not voice such feelings of concern and fear, and who appeared to give less importance to fever.

Despite the predominance of negative feelings in parents regarding childhood fever, certain non-health professional parents also experienced feelings of satisfaction and reassurance in the following cases: a) when the fever resolved (shared also by the health professional parents), b) when they had support for the care of their child, or c) when a health center was close at hand.

Parents who did not belong to the health sector demanded the need for more information to allow them to distinguish between a “high fever” and a normal temperature and to recognize possible complications of fever.

In relation to the sources of information consulted, the participants from both groups noted resorting to health professionals, medication information leaflets, parenting magazines and the internet. The internet was used by most participants, although certain parents were opposed to its use, others acknowledged selecting and screening the information available online. On the other hand, non-health professional parents also consulted pharmacists and other peer groups (friends and family members), highlighting the role of grandmothers as sources of “information overload”. Meanwhile, scientific publications and the use of applications for mobile phones or tablets (Apps) related to the subject were used as a source of information in the case of health professionals.

## 4. Discussion

To the best of our knowledge, this is the only study to analyze the conceptualizations of parents regarding fever, performing a comparison between parents who are health professionals and those who are not, and parents from both rural and urban environments. In addition, this is one of the few research studies to explore this phenomenon based on segmented FGs, providing a novel and comprehensive vision of the same, even though no differences were found among parents according to their place of residence.

Regarding the conceptualizations associated with the classification, cause, symptoms and evolution of fever, no substantial differences were found among participants according to their training. Thus, parents without health training included measures of fever prevention such as prophylaxis with probiotics and physical measures, besides immunization measures, which also appeared in parents who were health professionals. This study highlights feelings of greater responsibility and embarrassment when consulting colleagues in the case of parents who were health professionals, which are perceptions that influence their behavior in the search of information when their children experience episodes of fever.

Concerning the methods for fever detection used by parents, we encountered similarities with previous studies highlighting symptoms such as red cheeks, glassy eyes or changes in the child’s behavior as being indicative of fever [25,26,27,28,29]. Additionally, this study reports the parents’ belief that each child presents a certain singularity in their fever symptoms and a degree of tolerance towards fever, as acknowledged by the parents of both groups, which is valued as being good or bad.

Regarding the causes of fever perceived by parents, our results coincide with previous studies which include infections, a climatic influence, popular belief or chance [27]. This study also provides new causes identified by the parents of both groups such as weaning, the process of child growth, going to a nursery school or reactions to certain vaccines. Furthermore, non-health professional parents also mentioned tantrums or tiredness as possible causes of fever.

Along the lines of former studies which analyzed the perceptions of parents on methods for the prevention of fever, the parents in this study highlighted the importance of immunization [30]. Additionally, our results highlight the belief of non-health professional parents that prophylaxis with probiotics, an appropriate diet, or keeping the children warm, specifically by covering what they considered to be their ‘weak’ points (ears and throat) constitute measures of fever prevention.

Certain studies have focused on the self-limited nature of fever [29]. Parents identify this characteristic in the evolution of fever, highlighting the existence of fever variations throughout the day.

In this study, the parents’ sentiments regarding fever are consistent with other qualitative studies that suggest that parental anxiety increases when the child’s temperature is higher, when there is a possibility of suffering from a severe or potentially mortal illness, and in the case of fever complications (e.g., febrile convulsions). Furthermore, parental anxiety decreases as the children become older [27,31] because fever management is quite different if observed in children younger than three months taking guidelines from the National Institute for Clinical Excellence (NICE) into account [32]. A new finding in this study shares the anxious feelings perceived by the parents who had a healthcare background, who perceived that they had an extra responsibility regarding the care of their children due to their training. Besides, the fact that the health professional parents felt ashamed to consult fellow colleagues due to embarrassment or because of the social consideration that they had to know how to treat their children is another aspect which warrants consideration.

As reported in previous studies, the parents’ concerns regarding the symptoms of fever are greater the longer a high fever lasts, or in the case of febrile episodes that are resistant to antipyretics and when the fever increases at night [6,29], these concerns are inversely proportional to the number of children [33]. Moreover, this study discloses the following feelings acknowledged by parents regarding the presence of fever in their children: freezing up (unable to react, associated with feelings of guilt), being “bad parents” and experiencing an obsession with taking the child’s temperature.

Also, this study reveals that parents not only experience negative feelings regarding fever, but also, positive feelings of relief and reassurance accompany the resolution of the same. In the case of non-health professional parents, they also felt reassurance if they were close to a health center and when they had the support of someone else in the care of the child with fever.

Regarding the sources of information consulted in the event of febrile episodes, our study confirmed a tendency previously noted in studies considering the use of web 2.0 tools such as the internet and social media (such as WhatsApp among peers (friends, family)), as alternative sources of information besides health professional consultations [8,33]. Despite the relevance that these sources are acquiring amongst the population, we know that the information that they disclose may sometimes be lacking in solid scientific evidence. In this sense, we observed that within the participants’ discourses, certain criticisms were voiced regarding the information available online, questioning its quality, validity and consistency. Therefore, future research must continue to analyze the quality of the online information in order to provide improved health education guidance.

### Limitations

One of the possible limitations of the study is the memory bias of the participants due to the time elapsed since their PCES consultation and the performance of the FG. To avoid this, this time was limited to three months.

## 5. Conclusions

Parents’ perceptions of fever as being beneficial for their child’s health while at the same time being a damaging event, or as an event that may even lead to the death of the child, determines the approach of parents during fever episodes. 

The results of this study are useful for both health professionals and managers, as they help improve our understanding of how parents’ approach and perceive fever in children, thus helping to emphasize and not trivialize their feelings. Additionally, this study provides key information for the development of educational programs on childhood fever directed at managing parents’ anxiety and providing basic knowledge on the possible causes, symptoms and complications of fever.

## Figures and Tables

**Figure 1 ijerph-16-02959-f001:**
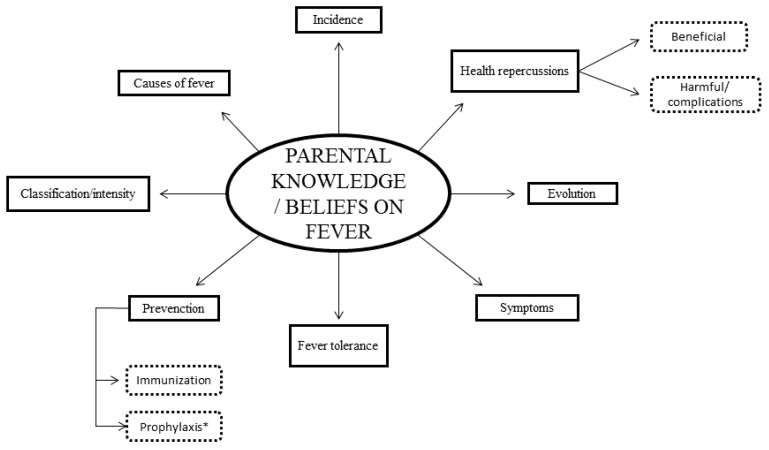
Knowledge and beliefs of parents on fever.

**Figure 2 ijerph-16-02959-f002:**
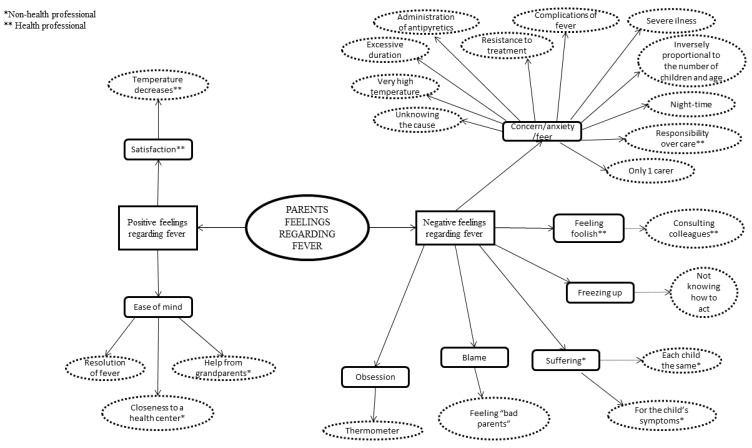
Parents’ ambivalent feelings regarding their child’s fever.

**Figure 3 ijerph-16-02959-f003:**
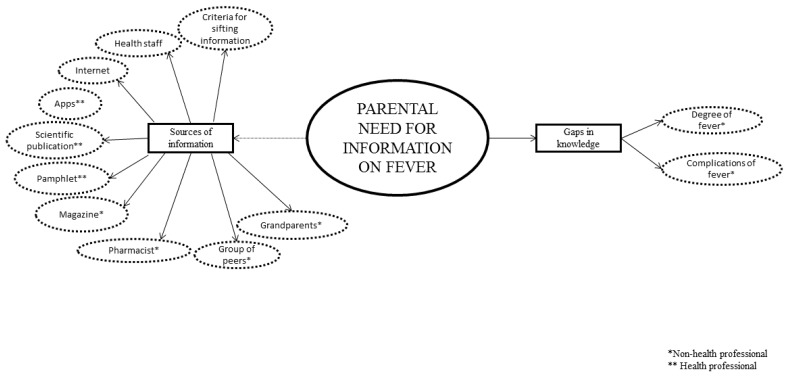
The need for information on fever perceived by parents.

**Table 1 ijerph-16-02959-t001:** Main characteristics of the parents participating in the focus groups.

Sex			Men	Women
Age	<30 years old		0	4
	30–40 years old		15	14
	>40 years old		12	12
Number of children	1		7	12
	2		11	13
	3–4		7	5
	>4		2	0
Place of residence (environment)	Rural environment		13	16
	Urban environment		14	14
Level of studies	No studies		0	0
	Primary studies		4	2
	Secondary studies		6	9
	University studies: diploma, Bachelors degree		16	18
	Master/PhD		1	1
Profession	Non-health professional		14	15
	Health professional	Medical degree	7	7
		Nursing diploma or degree	4	8
		Nurse aide/Health technician	2	0

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
