# Peer review of "A Qualitative Study of Parents’ Conceptualizations on Fever in Children Aged 0 to 12 Years"

_ijerph, 2019, doi:10.3390/ijerph16162959_

Round 1
Reviewer 1 Report
in essence, qualitative research is always 'different' from the much more commonly reported quantative research, but has its own methods. In my reading, these authors have respected these methods, so that the paper has its merits.
i only has some content related suggestions (taking guidelines like NICE into account, fever management is quite different if observed in young infants, < 90 days, and i completely miss this aspect in the paper. Secondly, there is a recent meta-analysis of Peetoom et al on the potential benefit of educaction of parents on the topic of fever, PIMD 27432451).
besides these content related suggestions, i recommend to improve the quality of the figures, since difficult to read in their current presentation.
Author Response
Response to Reviewer 1 Comments
Point 1: In essence, qualitative research is always 'different' from the much more commonly reported quantitative research, but has its own methods. In my reading, these authors have respected these methods, so that the paper has its merits.
Authors’ response 1: We thank the reviewer for this positive comment.
Point 2: I only has some content related suggestions (taking guidelines like NICE into account, fever management is quite different if observed in young infants, < 90 days, and I completely miss this aspect in the paper. Secondly, there is a recent meta-analysis of Peetoom et al on the potential benefit of education of parents on the topic of fever, PIMD 27432451)
Authors’ response 2: Thanks for your comments and recommendations. We have found this reference to be very interesting and useful. Therefore, we have now incorporated this study to the references.
Previous research has shown that a person’s educational level can determine their perceptions on the health-disease process, in which the academic level is inversely related to the amount of health-related knowledge and phobia regarding a range of symptoms [10-12].
(…)
Furthermore, parental anxiety decreases as the children become older [27,31] because fever management is quite different if observed in children younger than three months taking guidelines like NICE into account [32].
- Peetoom, K. K.; Smits, J. J.; Ploum, L. J.; Verbakel, J. Y.; Dinant, G. J.; Cals, J. W. Does well-child care education improve consultations and medication management for childhood fever and common infections? A systematic review. Archives of disease in childhood 2017, 102(3), 261-267, doi: 10.1136/archdischild-2016-311042
(…)
- NICE Guidance. Fever in under 5s: assessment and initial management. Available online: https://www.nice.org.uk/guidance/cg160 (accessed on 11 august 2019).
Point 3: Besides these content related suggestions, I recommend to improve the quality of the figures, since difficult to read in their current presentation.
Authors’ response 3: Thanks for your comment. We have improved the quality of the figures changing its size and the format (*png).
Reviewer 2 Report
See attached document below.

Author Response
Response to Reviewer 2 Comments
Point 1: The first sentence of the introduction cites an article to support the idea that parents tend to “consider fever a disease, instead of a symptom or sign of an illness.” The article cited does not support this statement. They should find a more appropriate citation or delete the first sentence of the Introduction.
Authors’ response 1: Thanks for this comment. We have revised the references, in fact it is a mistake and we have already corrected.
Parents have a certain tendency to consider fever as a disease, instead of a symptom or sign of illness [1].
- 1. Ertmann, R.K.; Reventlow, S.; Söderström, M. Is my child sick? Parent`s management of signs of illness and experiences of the medical encounter: Parents of recurrently sick children urge for more cooperation. Scand J Prim Health Care. 2011, 29,23-27, doi:10.3109/02813432.2010.531990
Point 2: In the Methods section, they do not indicate the number of parents that participated in the study in the narrative. This is not mentioned in the text until line 112 in the Results section. I suggest adding this information to the Methods section.
Authors’ response 2: We thank the reviewer for this comment. Following the reviewer’s recommendations, we have included this information in the Methods section. Please, see this section.
Point 3: The remainder of the issues deal with grammar or semantics:
- Line 47, the first sentence does not seem complete. I suggest adding “This was” to the beginning of the sentence.
- Line 57, they use the word carers (parents/carers). Carers refers to individuals caring
for disabled or elderly individuals that cannot care for themselves. I suggest they use
caregivers, instead of carers, here and throughout the manuscript.
- Line 115, I suggest using from instead of to in the sentence’ “The FGs of fathers took
place separately from (instead of to) those of mothers.
- Line 146, “whose” should be replaced with “those” at the end of that line.
- Line 150, I suggest eliminating the beginning of that sentence “According to their
health training the” and begin the sentence at the word “Parents”.
- Line 156, they include both arthralgia and joint aches which may be redundant.
- Line 185, they use the word manifested inappropriately. I suggest using “stated” instead of manifested.
- Line 202, replace manifested with “experienced.”
- Line 210, replace manifested with “noted.”
- Line 217, change “with” to “to.”
- Line 221, change “Besides” to “In addition.”
- Line 222, I suggest changing the beginning to that line to read “research studies to
explore”.
- Line 229, eliminate the word “Besides”.
- Line 230, change “ridicule” to “embarrassment” which is more consistent with what is stated elsewhere in the manuscript.
- Line 239, change “Besides” to “This study also”.
- Line 255, change “As a novelty” to “A new finding in”.
- Line 257, eliminate the word “Besides”.
- Line 267, eliminate the word “unpublished”
- Line 268, change the word “referred” to “felt”.
Authors’ response 3: We thank the reviewer for pointing us the suggestions for edits related to grammar and semantics. We have revised these aspects and we have edited as suggested (a-s).
- Line 47: This was a qualitative study based on grounded theory.
- We have revised the text in detail to change the use of caregivers, instead of carers.
- Line 76 instead of 115: “The FGs of fathers took place separately from those of mothers (i.e. these were separated by gender)”
- Line 135 instead of 146: “According to the intensity of the fever, both parents who were health professionals and those who were not, classified fever as: low-grade fever (or “normal fever” for some parents without health education), which was considered when the temperature was lower than 38ºC; high fever (up to 40ºC); and very high fever or “fever spikes” (40-41ºC).”
- Line 139 instead of 150: “Parents from both groups highlighted childhood “immunization” measures as methods for the prevention of fever” instead of “According to their health training, the parents from both groups highlighted childhood “immunization” measures as methods for the prevention of fever.”
- Line 143 instead of 156: “The signs and symptoms of fever mentioned by parents of both groups were: overall discomfort, arthralgia, headache, muscle aches” instead of “The signs and symptoms of fever mentioned by parents of both groups were: overall discomfort, arthralgia, headache, joint aches”
- Line 184 instead of 185: “Also, part of the parents of both groups stated that the presence of fever in their children caused them feelings of suffering and pity.”
instead of “Also, part of the parents of both groups manifested that the presence of fever in their children caused them feelings of suffering and pity.”
- Line 201 instead of 202: “Despite the predominance of negative feelings in parents regarding childhood fever, certain non-health professional parents also experienced feelings of satisfaction” instead of “Despite the predominance of negative feelings in parents regarding childhood fever, certain non-health professional parents also manifested feelings of satisfaction”
- Line 231 instead of 210: “In relation to the sources of information consulted, the participants from both groups noted resorting to health professionals” instead of “In relation to the sources of information consulted, the participants from both groups manifested resorting to health professionals”
- Line 238 instead of 217: “Meanwhile, scientific publications and the use of applications for mobile phones or tablets (Apps) related to the subject were used as a source of information in the case of health professionals.” instead of “Meanwhile, scientific publications and the use of applications for mobile phones or tablets (Apps) related with the subject were used as a source of information in the case of health professionals.”
- Line 281 instead of 221: “In addition, this is one of the few researches to study this phenomenon based on segmented FGs” instead of “Besides, this is one of the few researches to study this phenomenon based on segmented FGs”
- Line 282 instead of 222: “In addition, this is one of the few research studies to explore this phenomenon” instead of “Besides, this is one of the few researches to study this phenomenon”
- Line 288 instead of 229: we have deleted the word “Besides”.
- Line 290 instead of 230: “This study highlights feelings of greater responsibility and embarrassment when consulting colleagues in the case of parents who were health professionals” instead of “This study highlights feelings of greater responsibility and ridicule when consulting colleagues in the case of parents who were health professionals”
- Line 299 instead of 239: “This study also provides new causes identified by the parents of both groups such as weaning” instead of “Besides, this study provides new causes identified by the parents of both groups such as weaning”
- Line 316 instead of 255: “A new finding in, this study shares the anxious feelings perceived by the parents who had a healthcare background” instead of “As a novelty, this study shares the anxious feelings perceived by the parents who had a healthcare background”
- Line 318 instead of 257: we have deleted the word “Besides”.
- Line 329 instead of 267: we have deleted the word “unpublished”.
- Line 330 instead of 268: “they also felt reassurance if they were close to a health center and when they had the support of someone else in the care of the child with fever.” instead of “they also referred reassurance if they were close to a health center and when they had the support of someone else in the care of the child with fever.”